# muCool: muon cooling for high-brightness $\mu^+$ beams

**Aldo Antognini[1,2⋆] and David Taqqu[1,2]**

**1** Institute for Particle Physics and Astrophysics, ETH Zurich, 8093 Zurich, Switzerland
**2** Paul Scherrer Institute, 5232 Villigen–PSI, Switzerland

⋆ aldo@phys.ethz.ch

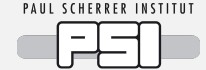

## Abstract

A number of experiments with muons are limited by the poor phase space quality of the muon beams currently available. The muCool project aims at developing a phase-space cooling method to transform a surface $\mu^+$ beam with 4 MeV energy and 1 cm size into a slow muon beam with eV energy and 1 mm size. In this process the phase space is reduced by a factor of $10^9 - 10^{10}$ with efficiencies of $2 \cdot 10^{-5} - 2 \cdot 10^{-4}$. The beam is then re-accelerated to keV-MeV energies. Such a beam opens up new avenues for research in fundamental particle physics with muons and muonium atoms as well as in the field of $\mu$SR spectroscopy.

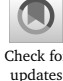
## 30.1 Introduction

Precision experiments with muons [1] require muon beams with large rates and low energy. Thus experiments often make use of secondary beam lines with large acceptance tuned to transport surface muons with momentum $p = 29$ MeV/c (equivalent to 4 MeV kinetic energy). These $\mu^+$ are copiously created by $\pi^+$ stopping close to the surface of the pion production target. Muons of lower momenta, from $\pi^+$ decaying below the surface of the target, can also be extracted from the production target. However, because of the momentum straggling in the target, the intensity of these sub-surface muon beams decreases rapidly with momentum ($p^{3.5}$-dependence [2]).

The large area of the production target, the scattering in the target, and the large acceptance of the secondary beamline result in muon beams with poor phase space quality ($\sigma_{x,y} \approx 10$ mm, $\theta_{x,y} \approx 100$ mrad) [3–5]. The muCool project aims to improve the phase-space quality of these secondary $\mu^+$ beams by a factor of $10^9 - 10^{10}$ while reducing the efficiency by only $2 \cdot 10^{-5} - 2 \cdot 10^{-4}$, transforming a standard secondary $\mu^+$ beam into a sub-mm keV beam.

## 30.2   The muCool compression scheme

In the proposed muCool scheme [6], a surface muon beam propagating in the $-z$-direction is slowed down in a He gas target featuring a strong electric ($E$) field inside a strong magnetic ($B$) field as shown in Figure 30.1. In the slowing-down process, the muon energy is rapidly reduced to the eV range where the E-field becomes important. The E-field, in conjunction with the B-field and gas density gradients, leads to drifting of the slowed-down muons drastically reducing their initially large spatial extent. In this drift process in the gas, the muons are guided into a sub-mm spot.

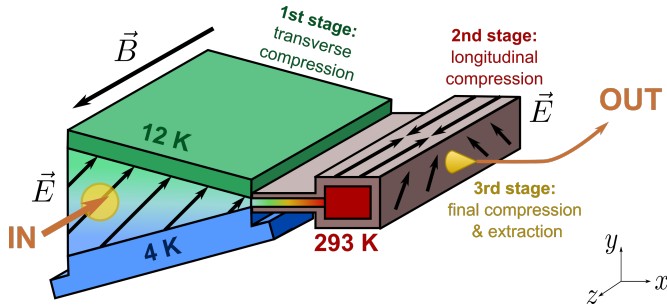

Figure 30.1:   Schematic diagram of the muCool device. A surface muon beam is stopped in a cryogenic He gas target with a vertical temperature gradient inside a 5 T field. The extent of the stopped muons is reduced first in the transverse ($y$), then in the longitudinal ($z$) direction using a complex arrangement of E-field and gas density gradient. The compressed muon beam is then extracted through an orifice into vacuum and re-accelerated along the $z$-axis.

The drift velocity of the $\mu^+$ in a gas with E- and B-fields is given by [7]

$$\vec{v}_D = \frac{\mu|\vec{E}|}{1+\omega^2/\nu^2}\left[\hat{E} + \frac{\omega}{\nu}\hat{E}\times\hat{B} + \frac{\omega^2}{\nu^2}\left(\hat{E}\cdot\hat{B}\right)\hat{B}\right]. \tag{30.1}$$

In this equation $\mu$ is the muon mobility, $\omega = eB/m$ the cyclotron frequency of the muon, $\nu$ the average $\mu^+$–He collision rate, and $\hat{E}$ and $\hat{B}$ the unit vectors of the electric and magnetic fields, respectively.

The spatial extent of the muon stop distribution decreases by making $\vec{v}_D$ position-dependent, so that $\mu^+$ stopped at different locations in the target drift in different directions, and converge to a small spot. This can be achieved by applying a complex E-field pointing in different directions at different positions, and by making the collision frequency $\nu$ position-dependent through a height-dependent gas density.

The muCool setup is conceived as a sequence of stages having various density and electric field conditions. In the first stage, which is at cryogenic temperatures, the muon beam is stopped and compressed in $y$-direction (transverse compression). In the second stage, which is at room temperature, the muon beam is compressed in $z$-direction (longitudinal direction). In the third stage, the muons are extracted from the gas target into vacuum, re-accelerated in $-z$-direction, and extracted from the B-field.

The 4-MeV $\mu^+$ beam with $\sigma_{x,y} \approx 10$ mm is degraded in a moderator and then stopped in the first stage of the muCool target containing the He gas at cryogenic temperatures and 10 mbar pressure. In this first stage, the third term in (30.1) is zero because $\vec{E} = (E_x, E_y, 0)$, with $E_x = E_y \approx 1$ kV/cm, is perpendicular to the B-field $\vec{B} = (0, 0, -|B|)$ and at 45° with respect to the $x$-axis. The peculiarity of this stage is the presence of a strong temperature gradient

in vertical direction from about 4 K to 12 K as shown in Figure 30.1. At lower densities (top part of the target) the collision frequency $\nu \approx 3$ GHz is smaller than the cyclotron frequency $\omega \approx 4$ GHz and therefore $\vec{v}_D$ is dominated by the $\hat{E} \times \hat{B}$ term in (30.1). Hence, the muons that are stopped in the top part of the target move downwards (in $-y$-direction) while drifting in $+x$-direction. By contrast, at larger densities (bottom part of the target) the collision frequency $\nu \approx 55$ GHz is larger than the cyclotron frequency $\omega$. Therefore, $\vec{v}_D$ in this region is dominated by the first term in (30.1), resulting in a drift velocity approximately in the $\hat{E}$ direction, so that muons stopped in the lower part of the target move upwards (in $+y$-direction) while drifting in $+x$-direction. Combining these considerations, we see that the first stage is used to stop the muons and to compress the vertical extension of the large stopping distribution.

The $\mu^+$ drifting in $x$-direction then enter into the second stage, which is at room temperature and has a field $\vec{E} = (0, E_y, \pm E_z)$, with $E_y = 2E_z = 0.1$ kV/cm, with a strong $z$-component pointing towards $z = 0$. Because $\nu$ is small at room temperature, the $\mu^+$ motion in this stage is dominated by the third term of (30.1) resulting in a fast reduction of the longitudinal extent. During this fast compression, the $E_y$-component (see $\hat{E} \times \hat{B}$ term in (30.1)) drifts the $\mu^+$ in $x$-direction towards the extraction stage. From there, the compressed beam can be extracted though a small orifice into vacuum, and moved quickly into a region of low gas pressure where re-acceleration can occur. Finally the beam needs to be extracted from the solenoid through an iron grid that terminates the magnetic field lines.

## 30.3 Demonstration of transverse and longitudinal compression

To demonstrate transverse compression, a cryogenic target as sketched in Figure 30.2 (left) was constructed, capable of sustaining the needed temperature gradient in vertical direction [8]. The target walls were lined with conducting tracks held at high voltage to define a homogeneous electric field at 45° angle w.r.t. the $x$-axis. A 13 MeV/c sub-surface muon beam was injected into the target and the slowed down muons drifted in the $x$-direction towards the tip of the target, while being compressed in $y$-direction by the combined action of the E- and B-fields, as well as the density gradient. A simulation of the $\mu^+$ trajectories is shown in Figure 30.2 (middle). To study the $\mu^+$ motion, a system of plastic scintillators detecting the positrons from muon decay was placed around the target. The recorded time spectra (see Figure 30.2 (right)) of these detectors were compared to simulations and good agreement was found [8, 9].

To test the longitudinal compression, a room temperature target as sketched in Figure 30.3 (left) has been constructed with a wall-lining defining E-field with components in $z$- and $y$-direction [10, 11]. A 10 MeV/c muon beam was injected and slowed down in the elongated target. Muons drifted towards the target mid-plane at $z = 0$ using the $z$-components of the E-field while the $y$-component drifted the $\mu^+$ in the $x$-direction. Such behavior is demonstrated by the simulated trajectories in Figure 30.3 (middle). A scintillator telescope (T1&T2), visible in Figure 30.3 (left) is used to measure the $\mu^+$ accumulating around $z = 0$. The measured time spectrum given in red in Figure 30.3 (right) shows that muons can be attracted in a short time to the $z = 0$ plane. Also in this case, good agreement between simulated and measured time spectra has been observed [11].

Summarizing, both transverse and longitudinal compression have been tested independently [9]. The observed time spectra for various experimental conditions behave as expected from simulations, validating the simulations, in particular the assumed cross sections obtained from scaling of proton data [9].

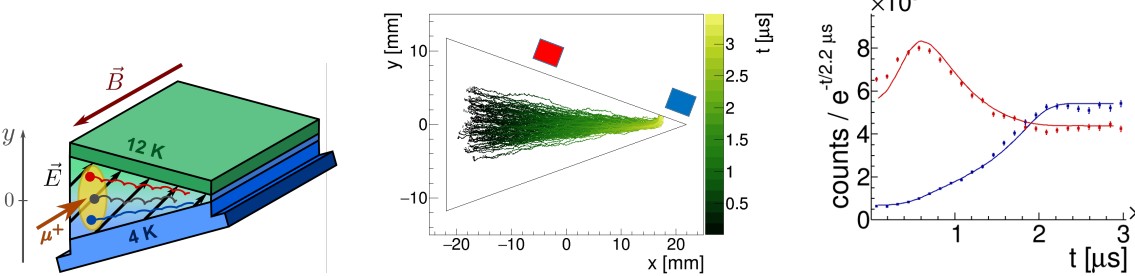

Figure 30.2: (Left) Sketch of the target used to test the transverse compression. (Middle) GEANT4 simulation of muon trajectories starting at $x \approx -15$ mm and drifting with time in $+x$-direction while compressing in the $y$-direction. The approximate positions of two plastic scintillators (red, blue) used to measure decay positrons are indicated. (Right) Measured and simulated time spectra for the two plastic scintillators indicated in the middle panel. The time zero is given by a counter detecting the muon entering the target. The counts are lifetime compensated, i.e., divided by $e^{-t/2.2\,\mu s}$.

## 30.4 Shortcut: the mixed transverse-longitudinal compression

According to the scheme of Figure 30.1 the next step would be to develop a target where the transverse compression stage is followed by a longitudinal compression stage. For this purpose, a connection between the cryogenic and the room temperature parts must be realized, with a short $\mu^+$ transit time. To avoid this challenge, a cryogenic target has been developed, in which both transverse and longitudinal compression occur simultaneously [9]. Such a mixed transverse-longitudinal compression target can be realized by adding a longitudinal component to the E-field of the transverse target (see Figure 30.4 (left)). The resulting $\mu^+$ motion in this target is sketched in Figure 30.4 (right).

Targets based on this concept have been simulated, developed and commissioned. The measured performance confirms the validity of this approach and of the simulations. They show that in the target a muon stop distribution with volume $\Delta x \times \Delta y \times \Delta z = 10 \times 10 \times 50$ mm$^3$ can be transformed within about $5\,\mu s$ into a beam drifting in $x$-direction in the He gas with 10 eV kinetic energy and capable of passing an aperture of $\Delta y \times \Delta z = 1 \times 1$ mm$^2$ size with efficiency larger that 50% (excluding muon decay losses).

The simplicity of this target and the shortening of the total (transverse + longitudinal) compression time is a major advantage of this configuration compared to the original proposal [6]. Its major downside is the shorter active region in $z-$direction which is limited by the time needed for the longitudinal compression at the much higher gas density compared to the scheme in the original proposal with longitudinal compression at room temperature.

## 30.5 Vacuum extraction and re-acceleration

The mixed-compression target can be modified to allow $\mu^+$ extraction from the gas target into "vacuum" through an orifice of about 1 mm diameter. To compensate for the He atoms leaving the target through the same orifice, new He gas has to be continuously injected into the system. We plan to inject the He gas right at the orifice, perpendicular to the $\mu^+$ motion (see Figure 30.5 (left)), so that the injected gas acts as a barrier for the target gas. The injected gas needs to be efficiently evacuated through a system of differentially pumped regions, so that the $\mu^+$ leaving the target experience a rapid decrease of the collision rates with the He atoms,

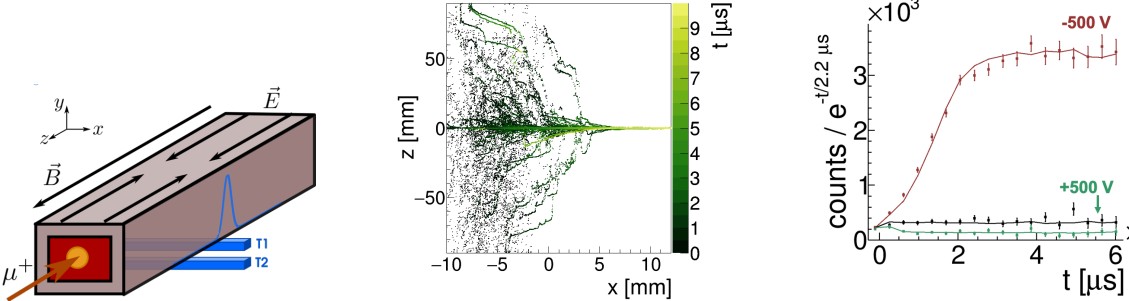

Figure 30.3: (Left) Sketch of the setup used to test the longitudinal compression. The scintillators T1 and T2 in coincidence constrain the $\mu^+$ accumulating to $z \approx 0$. The blue curve indicates the region of acceptance for coincident events. (Middle) Simulated $\mu^+$ trajectories. (Right) Measured and simulated time spectra for negative HV (red), positive HV (green) and no HV (black) at the target mid plane. The counts are lifetime compensated.

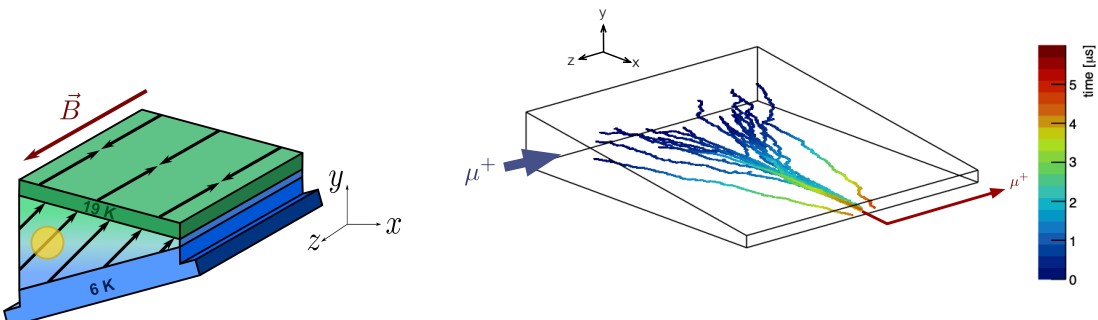

Figure 30.4: (Left) Sketch of the mixed transverse-longitudinal compression target with a vertical density gradient, $E_x$ and $E_y$ components as in the transverse compression target, and an $E_z$ component pointing to the target mid-plane at $z = 0$. (Right) Sketch of the muon trajectories in the mixed-compression target.

which is necessary to maintain a good beam quality.

We plan to define the electric field for the $\mu^+$ transport out of the target by adding six electrodes at the tip of the target as shown in Figure 30.5. The two electrodes at $z = 0$, connected to the high-voltages HV2 and HV5, define an E-field pointing in $y$-direction to drift the muons in $+x$-direction from the gas target to the re-acceleration region. The other two pairs of electrodes connected to the high-voltages HV3,6 and HV1,4 are kept at a slightly larger potential compared to HV2 and HV5 to define a V-shaped potential in $z$-direction with minimum at $z = 0$. This V-shaped potential confines the $\mu^+$ around $z \approx 0$ while they drift in $+x$-direction from the target to the re-acceleration region. A small electrode could be located in the re-acceleration region acting as a pulsed gate: for a short time it cancels one side of the V-shape potential barrier so that $\mu^+$ in the gate region can escape the confinement and be re-accelerated in $-z$-direction to a kinetic energy given by HV2 $\simeq$ HV5 $\approx$ 10 kV. Switching of this gate-electrode with high repetition rates up to about 1 MHz is needed to minimize losses in the accumulation and re-acceleration processes. Alternatively, the muon could be re-accelerated in $-z$-direction in a continuous way (without any pulsed gate) simply by modifying the six electrodes so that at a position along the x-axis (with sufficiently good vacuum conditions)

one side of the V-shaped confinement (in $-z$-direction) is absent.

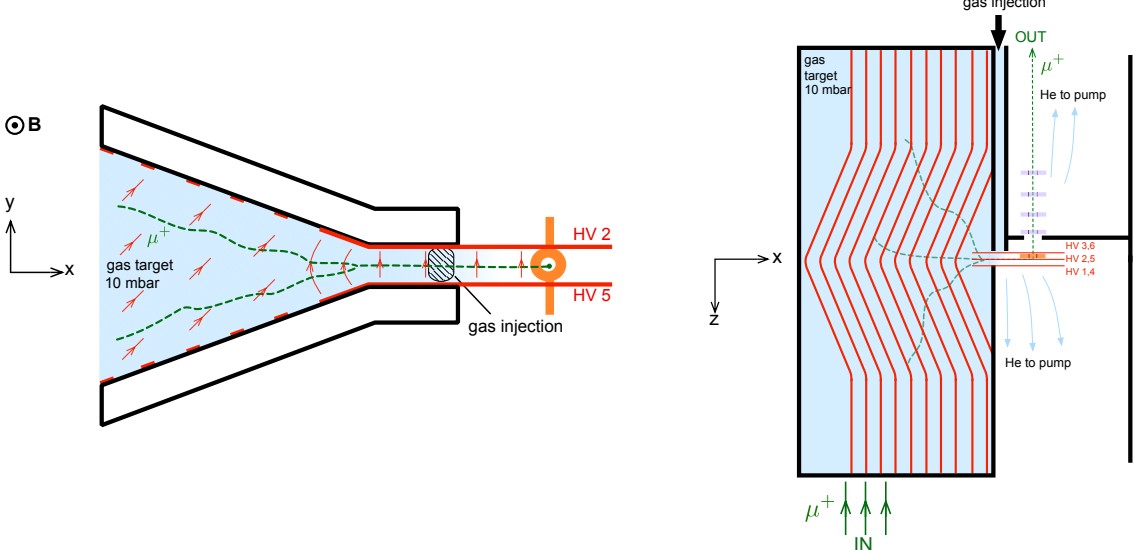

Figure 30.5: Schematic diagram (not to scale) of the baseline setup performing mixed longitudinal-transverse compression followed by vacuum extraction and re-acceleration. The electrodes are in red, the He gas flow is indicated by blue arrows, and muon trajectories are sketched in green. (Left) In the $xy$-plane the compression in vertical direction and the drift in $x$-direction is visible. In the orifice region, an E-field in $y$-direction, defined by the electrodes HV2 and HV5, is used to extract the muons from the gas target and to guide them in $+x$-direction due to the $\vec{E} \times \vec{B}$ drift. The electrode shown in orange is used as a pulsed gate to accelerate the muons. (Right) Similar to left panel, but for the $xz$-plane where the longitudinal compression and the re-acceleration in $-z$ direction are well visible.

## 30.6   The new beam

The muCool target transforms an input beam of 4 MeV energy beam, 1 cm diameter, 100 mrad divergence and 500 keV energy spread into a beam moving in He gas with 10 eV energy and about 1 mm diameter. Using the already-commissioned mixed-compression target, this transformation occurs within $5\,\mu$s with efficiencies of 90%, for a target with an active region of 50 mm length. This observed performance can be used to estimate the total conversion efficiency from muons entering the He gas target at 4 MeV energy to muons exiting the B-field of the solenoid with a kinetic energy of about 10 keV. Several other losses occurring prior or after the muon compression in the gas target have to be included as summarized in Table 30.1. As can be seen from this table, the total baseline (that assumes the commissioned gas target as a reference point) compression efficiency is estimated to be $7.5 \cdot 10^{-5}$. Note that this efficiency does not include possible losses at the incoupling of the solenoid and also does not account for the transverse (geometrical) acceptance of the gas target.

A muCool setup with this baseline efficiency applied to the $\pi E5$ beamline delivering surface $\mu^+$ at a rate of $2.1 \cdot 10^8\,\text{s}^{-1}$ (for the "slanted" target and 2.0 mA proton current), can yield a keV-energy beam with a rate of $2 \cdot 10^4\,\text{s}^{-1}$ and small phase space (40 mm mrad at 10 keV). Here, we assumed 25% incoupling losses due to reflections at the solenoid and transverse acceptance of the muCool target. Applied to the envisioned High Intensity Muon Beam HiMB, delivering $\mu^+$ with a rate of $1 \cdot 10^{10}\,\text{s}^{-1}$, a muCool output rate of $3 \cdot 10^5\,\text{s}^{-1}$ could be reached, provided

Table 30.1: Estimate of the muCool baseline efficiency using the commissioned mixed-compression target as a reference point for the compression towards the orifice. We thus assume here a target having an active region of 50 mm length operated at 10 mbar pressure with a 6-20 K temperature gradient. The stopping probability of 0.6% has been simulated assuming a surface muon beam with 10% (FWHM) momentum bite. A 3% (FWHM) momentum bite would increase the stopping probability to 1.6%. All the other entries have only been estimated and depend strongly on the upcoming R&D results.

| Efficiency | |
|---|---|
| $6 \cdot 10^{-3}$ | Stopping probability in He gas within the active region of the target |
| $1 \cdot 10^{-1}$ | Compression towards the orifice including muon decay losses (within 5 $\mu$s) |
| $6 \cdot 10^{-1}$ | Extraction from the orifice |
| $4 \cdot 10^{-1}$ | Drift from orifice to re-acceleration region (in about 2 $\mu$s) |
| $8 \cdot 10^{-1}$ | Muon decay from re-acceleration region to iron grid |
| $7 \cdot 10^{-1}$ | Transmission through iron grid terminating the B-field |
| $7.5 \cdot 10^{-5}$ | Total baseline compression efficiency |

the operational stability of the target is not disrupted by the higher degree of ionization of the high intensity muon beam (we assume 60% in-coupling losses).

The above-described baseline compression efficiency can be improved by extending the active region in $z$-direction, by increasing the longitudinal E-field strength, and by decreasing the gas temperature. At the cost of additional complexity, the stopping probability can be greatly increased by using a target with multiple active regions in $z$-direction, each having its own extraction orifice. In this case, the various beams exiting the target at different $z$-positions but same $x$- and $y$-positions, can be merged in the re-acceleration process into a single beam. The original scheme of Figure 30.1 can also be used to significantly extend the active region in $z$-direction. A moderate increase by a factor of 2 of the baseline efficiency would result in competitive beam rates of $4 \cdot 10^4$ s$^{-1}$ and $5 \cdot 10^5$ s$^{-1}$ when applying the muCool setup to the $\pi E5$ and the HiMB, respectively.

## 30.7 Selected possible applications

This new beam opens the way for next generation experiments with muons where the reduced phase-space is of great advantage.

The search for a muon EDM represents a well motivated channel for physics beyond the Standard Model [12]. While muon EDM searches with a sensitivity of $10^{-21}$ e cm are ongoing at Fermilab and J-PARC as a "by-product" of their efforts to measure the muon g-2 [13], a muon EDM experiment has been proposed at PSI based on a frozen-spin technique applied to a compact muon storage ring [14]. Preliminary studies show that a sensitivity of $6 \cdot 10^{-23}$ e cm could be reached in the PSI experiment using the $\mu E1$ beam at 125 MeV/c delivering $2 \cdot 10^8$ $\mu^+$/s. Because of the small phase space acceptance of the storage ring, the coupling efficiency for the $\mu E1$ beam is only $2.5 \cdot 10^{-4}$ so that only $5 \cdot 10^4$ $\mu^+$/s are stored in orbit. The muCool beam with a rate $5 \cdot 10^5$ s$^{-1}$ accelerated to 125 MeV/c or 200 MeV/c would result in a larger rate of stored muons as it avoids the coupling losses into the storage ring due to its small phase space.

The muCool beam can also greatly improve $\mu$SR investigations of sub-mm samples. Because the pile-up effects in the typically 10 $\mu$s-long observation time window become increas-

ingly unsustainable for rates exceeding $5 \cdot 10^4$ s$^{-1}$, the full HiMB-muCool potential could be exploited by switching the keV-energy sub-mm beam between several $\mu$SR instruments operating simultaneously.

Muon to vacuum-muonium conversion is very efficient for keV-energy muons [15]. Hence, the sub-mm muCool beam at keV-energy could be converted into a high-brightness muonium source. This novel muonium source could be exploited to improve on the precision of muonium spectroscopy by orders of magnitude (e.g. the 1S-2S with a relative accuracy of $10^{-12}$ [16]), and could be used to study the influence of gravity on the muonium to investigate the gravitational interaction of antimatter and second generation leptons in the earth's gravitational field [17].

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
