# Peer review of "muCool: muon cooling for high-brightness μ+ beams"

_SciPost Physics Proceedings, doi:SciPost Phys. Proc. 5, 030 (2021)_

## Round 1 · Referee Report · Adrian Signer (Referee 1) · 2021-6-17

Report

We (the editors Cy Hoffman, Klaus Kirch, Adrian Signer) had the
opportunity to review an earlier draft of the article and were in
communication with the authors before the submission. All our comments
and suggestions have been taken into account. Hence, we think the
paper can now be published in the current form.

---

## Round 1 · Referee Report · Anonymous (Referee 2) · 2021-6-28

Report

The paper reports on an innovative concept for compressing the phase space of the currently best available mu+ beams by a factor of 1E9-1E10, essentially transforming an MeV and cm sized beam into a keV and sub-mm sized compressed beam. Such a beam could enable a new generation of fundamental and applied physics experiments. This is a major undertaking which not only requires creative ideas, but years of effort of a technically superb and dedicated team to realize.

The concepts and demonstrated stages are clearly described, and estimates and
an outlook of the final beam properties and its physics potential are given.

As all the required aspects regarding the quality of the paper are clearly met, I recommend the publication of this work.

Requested changes

I have no requested changes, but added a few suggestions for consideration by the authors.

30.2 compression scheme.
For me it would be helpful if some typical numbers are given for
$\omega$, $\nu$ and the He densities at 4 and 12 K, so that the interplay between the different regimes in eq.30.1 becomes evident.

line 81: track -> traces?
electric field -> homogeneous electric field?

Fig. 30.3 Though shown on the ordinate, perhaps mention explicitly that muon decay has been divided out

line 120. Please explain why z length is limited in the combined scheme.

30.5 It would be helpful to separate more clearly between demonstrated steps and additional steps still under active R&D in the overall project.

table 30.1 Is
Is muon decay accounted for in the first 5 us?

---

## Round 2 · Author Response

Dear Editor

we agree with all the comments of the referee. All of them have been taken into account in this resubmission.

Many thanks and best regards
Aldo Antognini

---

## Round 2 · List of Changes

1) Referee:
For me it would be helpful if some typical numbers are given for ω, ν and the He densities at 4 and 12 K, so that the interplay between the different regimes in eq.30.1 becomes evident.

Our reply:
We inserted typical values of the requested frequencies. See lines 60-64.
"At lower densities (top part of the target) the collision frequency ν ≈ 3 GHz is smaller than the cyclotron frequency ω ≈ 4 GHz and therefore vD is dominated by the Eˆ × Bˆ term in (30.1). Hence, the muons that are stopped in the top part of the target move downwards (in −y-direction) while drifting in +x-direction. By contrast, at larger densities (bottom part of the target) the collision frequency ν ≈ 55 GHz is larger than the cyclotron frequency ω."

2) Referee
line 81: track -> traces?

Our reply:
Both terms are correct.

3) Referee:
electric field -> homogeneous electric field?

Our reply:
We have inserted "homogeneous". See line 81.

4) Referee:
Fig. 30.3 Though shown on the ordinate, perhaps mention explicitly that muon decay has been divided out.

Our reply:
We have inserted explicitly a comment that the counts are lifetime compensated. See last sentences of figures 30.3 and 30.2 captions.

5) Referee:
line 120. Please explain why z length is limited in the combined scheme.

Our reply:
We inserted following short explanation at lines 121-123.
"Its major downside is the shorter active region in z−direction which is limited by the time needed for the longitudinal compression at the much higher gas density compared to the scheme in the original proposal with longitudinal compression at room temperature."

6) Referee:
30.5 It would be helpful to separate more clearly between demonstrated steps and additional steps still under active R&D in the overall project.

Our reply:
All the steps described in Section 30.5 are active R&D and not yet demonstrated. So following the advice of the referee we have slightly modified the text accordingly:
- Line 133: "We plan to define...."
- Line 140: "A small electrode could be located..."
- Line 145: "Alternatively, the muon could be re-accelerated..."

7) Referee:
table 30.1. Is muon decay accounted for in the first 5 us?

Our reply:
Yes. To avoid possible doubts we specify more clearly the second entry of the table as
"Compression towards the orifice including muon decay losses (within 5 μs)"

---

## Editorial Decision

published